# Screening and Selection for Herbicide Tolerance among Diverse Tomato Germplasms

**Gourav Sharma** [1,2], **Swati Shrestha** [2,3,*] , **Te-Ming Tseng** [2] **and Sanju Shrestha** [3]

1. Bayer Crop Sciences, Saint Louis, MO 63017, USA
2. Department of Plant and Soil Sciences, Mississippi State University, Starkville, MS 39762, USA
3. Department of Plant and Soil Sciences, Oklahoma State University, Stillwater, OK 74078, USA
* Correspondence: swati.shrestha@okstate.edu

**Abstract:** *Solanum lycopersicum*, the domesticated species of tomato, is produced and consumed globally. It is one of the most economically important vegetable crops worldwide. In the commercial production of tomatoes, tomatoes are extremely sensitive to herbicide drifts from row crops in the vicinity. Injury to tomatoes from auxin herbicides and glyphosate can occur at rates as low as $0.01\times$. This results in a substantial yield reduction, and at high drift rates, plants may not show signs of recovery. With the new herbicide-resistant crop technologies on the market, which include 2,4-D and dicamba-resistant crops, there is an increase in the usage of these herbicides, causing more serious drift problems. There is a diverse germplasm of tomatoes that includes wild relatives which are tolerant to numerous biotic and abiotic stresses. Herbicide/chemical stress is an abiotic stress, and wild tomato accessions may have a natural tolerance to herbicides and other abiotic stresses. In the current study, diverse tomato genotypes consisting of 110 accessions representing numerous species, *Solanum habrochaites*, *S. cheesmaniae*, *S. pimpinellifolium*, *S. chilense*, *S. lycopersicum*, *S. pimpinellifolium*, *S. galapagense*, *S. chimelewskii*, *S. corneliomulleri*, *S. neorickii*, and *S. lycopersicoides*, were used for screening drift rate herbicide tolerance. The herbicides tested included simulated drift rates of 2,4-D, dicamba, glyphosate, quinclorac, aminopyralid, aminocyclopyrachlor, and picloram. The visual injury rating of each accession for each herbicide treatment was taken 7, 14, 21, and 28 days after treatment (DAT) on a scale of 0–100%. Numerous accessions were found to have minimal injury (less than 20%) for each of the herbicides tested; nine accessions were found for both 2,4-D and glyphosate, eleven for dicamba, five for quinclorac, eight for aminocyclopyrachlor and two for both aminopyralid and picloram at 28 DAT. The identification of genotypes with a higher herbicide tolerance will provide valuable genetic resources for the development of elite tomato varieties that can resist herbicide injury and produce competitive yields.

**Keywords:** herbicide-tolerant tomatoes; drift; genotypes

## 1. Introduction

Cultivated tomato (*Solanum lycopersicum*) has established itself as an indispensable agricultural commodity, a symbol of American cuisine, and a key driver of the country's economy. In 2022, annual US tomato production was approximately 11,423,714,520 kg, encompassing both fresh market and processing tomatoes, and providing a total value of about USD 1.7 billion [1]. Tomatoes are widely known for their outstanding antioxidant content, including their often-rich concentration of lycopene [2,3]. In Mississippi, the value of tomato production totals approximately USD 1.5 million. Even though the crop is primarily grown in the plasticulture system in the state, weeds are still a major problem in tomato production [4]. Major weeds in tomato cultivation are yellow nutsedge (*Cyperus esculentus*), purple nutsedge (*Cyperus rotundus*), large crabgrass (*Digitaria sanguinalis*), and palmer amaranth (*Amaranthus palmeri*). Among these weeds, yellow and purple nutsedge are the most problematic, causing significant yield losses and decreased fruit quality [5].

The herbicide options for tomatoes are limited, and only a few are highly effective on nutsedge. Herbicides registered for tomatoes for nutsedge control include halosulfuron, S-metolachlor, and trifloxysulfuron. Numerous studies have well established that, although the significant control of nutsedge and other weeds is achieved (60–90%) by the labeled herbicides, significant injury (15–54% injury) is also observed in tomato plants because of herbicide sensitivity [6,7]. Moreover, injury from herbicides drifting to greenhouse tomatoes from the neighboring fields leads to deformed fruits and yield reduction [8].

Off-site herbicide drift is devastating to vegetable producers, and negatively impacts humans, soil, and environmental health. For instance, in 2013 a small organic tomato grower in Tupelo, Mississippi, USA lost USD 22,550 due to the 2,4-D drift in his field [9]. Due to crop technologies such as glyphosate-resistant corn, soybean, and cotton, growers have primarily depended on glyphosate for weed control [10]. And, with the recent commercialization of 2,4-D-resistant corn, soybean, and cotton by Dow AgroSciences and dicamba-resistance crops by Monsanto, the use of auxin herbicides will increase significantly, thus resulting in a greater risk of drift of these herbicides to tomato fields. In 2014, USDA approved the commercialization of 2,4-D-tolerant corn from Dow AgroSciences (The Canadian Biotechnology Action Network, Nova Scotia, Canada, B3M 4H4), and then in 2015, the genetically engineered dicamba-tolerant soybean and cotton from Monsanto were approved for seed sale. With these technologies, the use of 2,4-D in corn is estimated to be increased by 30 times [11]. Glyphosate-resistant weeds have been reported in all the major US row crops, including soybean, corn, cotton, and wheat, and the best option to control these weeds will be using 2,4-D or dicamba, which are commonly used as post-treatments for glyphosate-resistant broadleaf weeds [12]. Thus, with these technologies, growers can apply the herbicides on labeled crops to achieve better weed control. Still, on the other hand, it could be a problem for sensitive non-target vegetables, organic growers, and rural home gardens due to possible drift.

According to Caseley and Coupland (1985) and Ovidi (2001), drifted rates of glyphosate can cause the shortening of pollen tubes, change in the shape of generative cells from spindle-like to elongated cylinder-like, absence of microtubules, malformations in reproductive organs, and delay in fruit ripening [13,14]. In 1974, Jordan and Romanowski reported that tomato plants sprayed with dicamba at the early bloom stage had significantly higher yield losses and scattered fruit sets [15]. Tomatoes are, therefore, very susceptible to herbicides like dicamba and glyphosate. Quinclorac is another synthetic auxin herbicide commonly used to control barnyardgrass in rice, and is the only auxin herbicide with grass activity [16]. However, tomato plants are very sensitive to quinclorac [17,18]. In Arkansas and Mississippi Delta, quinclorac is sprayed aerially, causing high drift to tomato fields [19]. The most common symptoms of quinclorac drift are severe leaf curling and cupping, small plant size, lack of vigor, bloom abscission, and low fruit yield [8]. A drift rate of quinclorac above 0.42 g/ha has the potential to reduce tomato yield and cause significant injury to the plants. Aminopyralid is a relatively new synthetic auxin herbicide, and is used only in permanent grass pastures and grass hay fields. There are no drift studies that have been conducted on tomato with this herbicide, but a study by Flessner et al. (2012) reported that aminopyralid causes a higher reduction in dry biomass and height in cotton compared to 2,4-D [20]. Aminocyclopyrachlor (AMCP) is a pyrimidine carboxylic acid-type herbicide; in simulated spray drift of aminocyclopyrachlor to flue-cured tobacco at five different rates from 0.31 g ae ha$^{-1}$ to 31.4 g ae ha$^{-1}$, plant injury increased from 11% to 77% (8 WAT) as the rate increased [21]. Picloram (4-amino-3, 5, 6-trichloro-2-pyridinecarboxylic acid) is an acidic herbicide in the pyridine carboxylic acid family used to control annual and perennial dicot weeds, shrubs, and woody vegetation. Smith and Geronimo reported a high sensitivity of tomatoes to picloram herbicides, which caused a significant yield loss at 11.2 g/ha in the field-grown tomatoes [22].

Breeding herbicide tolerance in tomatoes would be the most economical, environmentally friendly, and feasible method to protect tomatoes from drift injury and sensitivity to

labeled herbicides. In the current study, diverse tomato germplasm and its wild species were assessed for tolerance to various herbicides.

The objective of the study was to screen tomato germplasm for tolerance to herbicides that can potentially drift from the neighboring row crop fields. The results from this study could provide valuable resources for breeding herbicide tolerance traits into the agronomically important tomato varieties, thus ultimately allowing the growers access to tomato varieties with higher herbicide tolerance in the future and preventing yield reduction due to drift injury and weeds.

## 2. Materials and Methods

A collection of 107 wild/abiotic/biotic stress-tolerant tomato accessions was provided by the Tomato Genetic Resource Center at the University of California, Davis, CA, USA. Additionally, two accessions (Money Maker and Bonnie Best) were obtained from USDA in Geneva, New York, NY, USA, and eight cultivars (06 heat and 02 drought stress tolerant) were purchased from a commercial seed company (Seedman.com®, Gautier, MS, USA). A list of all the accessions used in the study is provided in Supplementary Table S1. To improve the germination of the seeds, they were treated for 10 min with 10% bleach solution, rinsed 5–6 times with sterile distilled water at room temperature, and then kept in sterile distilled water overnight at 4 °C to allow the seeds to imbibe water. Imbibed seeds were then planted into cone-tainers (Greenhouse Megastore, Danville, IL, USA) with a diameter of 3.81 cm and a depth of 20.95 cm, filled with Sungro professional growing mix, (Sungro Horticulture®, Agawam, MA, USA) and maintained in a greenhouse set at 23 °C for both day and night; light duration was set to 14 h. Cone-tainers were placed in 7 by 14 cone-tainer trays measuring $51 \times 30 \times 17$ cm. Tomato seeds were sown in a completely randomized design with four replications and two runs. Each replication consisted of six tomato plants. At the 4-leaf stage (a month after sowing), plants were treated with simulated drift rates of 2,4-D, glyphosate, dicamba, quinclorac, aminopyralid, aminocyclopyrachlor, and picloram in a spray chamber equipped with the TP8002EVS Even Flat Spray Tip (TeeJet®, Spraying Systems Co. World Headquarters, P.O. Box 7900, Wheaton, IL 60187, USA), calibrated to a deliver 186 L ha$^{-1}$ at 275.79 KPa while maintaining the constant speed of 4.8 Km/h. Drift rates were selected based on previous studies, and varied from 0.01X to 0.05X, where X is the recommended dose of herbicide [23]. Most of the drift rates are lower than the herbicide application rate, so a preliminary experiment was conducted to select herbicide rates in the range of 0.01X to 0.05X that would cause plant injury but not death of the plants. Based on a preliminary study, 2,4-D, dicamba, and glyphosate were applied at 0.01X of the recommended rate; quinclorac was applied at 0.01X; and aminopyralid, aminocyclopyrachlor, and picloram were applied at 0.05X rate [23–26]. Table 1 lists all the herbicides used in the study, along with their simulated drift rates.

**Table 1.** List of herbicides and their drift rates used in the study.

| Herbicide | Trade Name | Rate Used | Drift Rates (g ae ha$^{-1}$) |
|---|---|---|---|
| 2,4-D | Weedar-64® | 0.01X | 11.2 |
| Dicamba | Clarity® | 0.01X | 2.8 |
| Glyphosate | Roundup Powermax® | 0.01X | 8.4 |
| Quinclorac | Facet L® | 0.01X | 39.2 |
| Aminopyralid | Milestone® | 0.05X | 6.15 |
| Aminocyclopyrachlor | Streamline® | 0.05X | 15.65 |
| Picloram | Tordon® | 0.05X | 28.0 |

The visual injury was recorded at 7, 14, 21, and 28 days after treatment (DAT) on a scale of 0–100%, where 0% indicates no injury and 100% indicates death of the plant (Table 2). Accessions showing injury less than or equal to 20% were classified as tolerant accessions. A similar setup was used for screening all the herbicides.

**Table 2.** Tomato visual injury and its associated symptomology.

| Tomato Injury (%) | Tomato Symptomology |
|---|---|
| 0–10 | No symptoms to slight injury, no growth reduction |
| 10–30 | Slight-to-moderate injury. Slight growth reduction |
| 30–50 | Epinastic and twisting of leaves in auxin herbicides, slight-to-moderate growth reduction, white/yellow discoloration at the base |
| 50–70 | Moderate-to-severe injury, callusing on the stems in auxins, and growth reduction |
| 70–95 | Severe injury and no growth |
| 95–100 | Near to death or death |

Data were subjected to analysis of variance (ANOVA) and means were separated using Fischer's protected LSD test at P = 0.05 in the statistical program JMP® (Statistical Discovery™, from SAS). There was no difference in mean injury between the two runs in all the herbicide treatments, so data from both the runs were pulled together for analysis.

The ANOVA model used in this experiment is defined as Yi = μ + αi + ei, where Yi is the response variable, which includes injury of tomato accessions, μ is mean of response variable, alpha is the treatment effect on the acquisitions, and ei is the error ei~N(0, σ2), all of which are independently identical distributed.

## 3. Results and Discussion

The tomato accessions were classified as tolerant if plants showed injury less than or equal to 20% at 28 DAT (Figure 1). Analysis of variance indicated that injury was significantly different among accessions for 2,4-D, picloram, dicamba, quinclorac, and glyphosate (Table 3). However, aminocyclopyrachlor and aminopyralid did not show any significant difference in injury; the *p*-values for both were 0.2912 and 0.1155, respectively. Even though there was no significant difference among accessions in response to aminocyclopyrachlor and aminopyralid injury, some accessions showed injury of less than 20% in both of these groups, which is reported here as it could be a valuable resource for the tomato breeding community.

**Table 3.** The F ratio, sum of squares, and mean square for all tested herbicides, along with their significance, the Prob > F.

| Herbicide | Sum of Squares | Mean Square | F Ratio | Prob > F |
|---|---|---|---|---|
| 2,4-D | 97,293.27 | 1201.15 | 7.5987 | <0.0001 |
| Aminocyclopyrachlor | 93,162.80 | 970.446 | 1.1362 | 0.2912 |
| Aminopyralid | 66,377.215 | 677.319 | 1.3343 | 0.1155 |
| Dicamba | 56,952.692 | 720.920 | 1.5689 | 0.0445 |
| Glyphosate | 39,430.523 | 788.610 | 2.4212 | 0.0036 |
| Picloram | 93,820.05 | 1054.16 | 2.6679 | <0.0001 |
| Quinclorac | 43,175.610 | 881.135 | 2.8772 | 0.0011 |

The order of the severity of the herbicide on different accessions was calculated in the one-way analysis with herbicide and injury, and indicates picloram to be the most injurious, whereas dicamba is the least injurious to tomato (Figure 2). The order of the herbicide injury in descending order of severity was picloram > aminopyralid > quinclorac > aminocyclopyrachlor > 2,4-D > glyphosate > dicamba. Wax et al. (1969) studied the drift of picloram, 2,4-D, and dicamba on soybean, and concluded that picloram is more injurious than the other two herbicides [27]. Flessner et al. (2012) reported that aminocyclopyrachlor is more injurious than 2,4-D in a study comparing drift of aminocyclopyrachlor and 2,4-D on cantaloupes, eggplant, and cotton, which is similar to our results, thus indicating plants are more sensitive naturally to aminocyclopyrachlor [20].

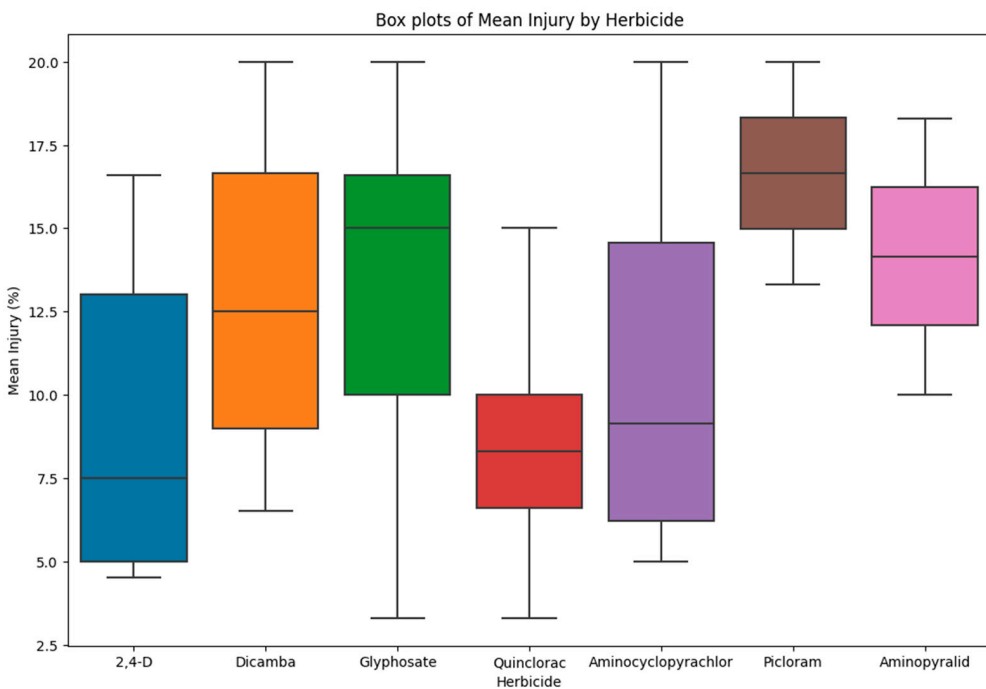

**Figure 1.** Box plot showing mean injury of tolerant accessions to different herbicides used in the study.

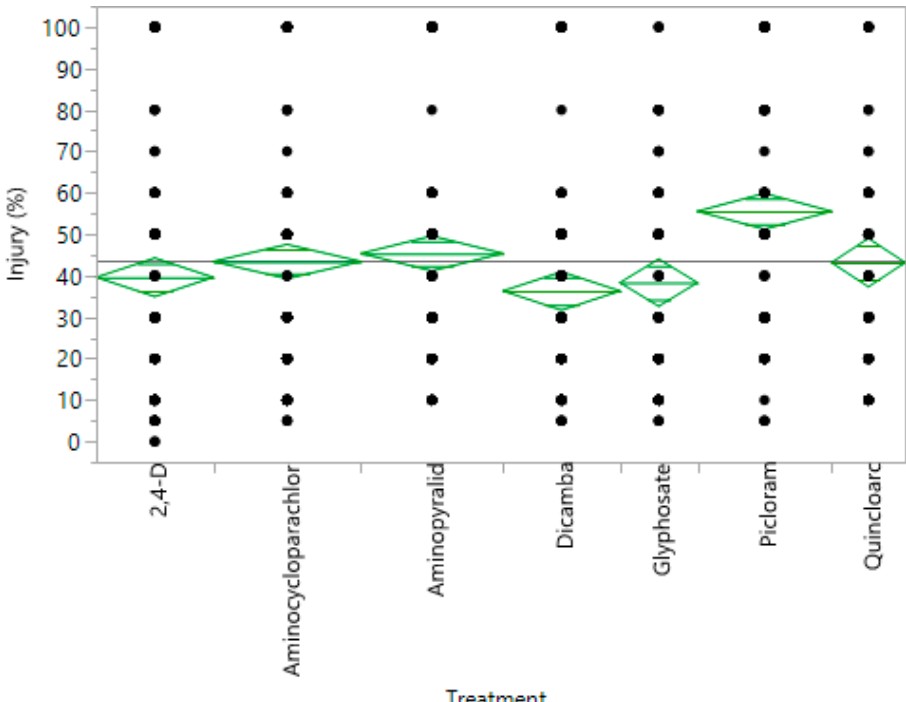

**Figure 2.** Comparison of injury ratings for all the herbicides. Presenting the most and least injured plant for each herbicide treatment. Black dots represent all the different values of injury for their respective herbicide. The top point on the green diamond is the upper confidence interval, whereas the lower point is the lower confidence interval. The black line gives the average injury rating for all herbicides combined.

Nine of the accessions were found to have a higher tolerance to 2,4-D; injury for tolerant accessions ranged from 5 to 20% (Table 4).

**Table 4.** List of tolerant accessions for each herbicide; all have injury less than or equal to 20%.

| Herbicide | Accession | Mean Injury (%) (28 DAT) |
|---|---|---|
| 2,4-D | TOM45 | 7.5J * |
| 2,4-D | TOM1 | 7.5J |
| 2,4-D | TOM56 | 5J |
| 2,4-D | TOM11 | 16.6HIJ |
| 2,4-D | TOM13 | 14.5HIJ |
| 2,4-D | TOM14 | 13IJ |
| 2,4-D | TOM22 | 11.6IJ |
| 2,4-D | TOM83 | 5J |
| 2,4-D | TOM17 | 4.5J |
| Dicamba | TOM1 | 8.5G |
| Dicamba | TOM3 | 9.5FG |
| Dicamba | TOM35 | 20EFG |
| Dicamba | TOM18 | 18.3EFG |
| Dicamba | TOM74 | 20EFG |
| Dicamba | TOM13 | 7.3FG |
| Dicamba | TOM14 | 15FG |
| Dicamba | TOM17 | 6.5G |
| Dicamba | TOM12 | 13.6FG |
| Dicamba | TOM262 | 12.5FG |
| Dicamba | TOM44 | 11.6FG |
| Glyphosate | TOM46 | 10.0IJ |
| Glyphosate | TOM60 | 3.3J |
| Glyphosate | TOM61 | 10.0IJ |
| Glyphosate | TOM64 | 11.6IJ |
| Glyphosate | TOM108 | 16.6HIJ |
| Glyphosate | TOM66 | 18.3GHIJ |
| Glyphosate | TOM18 | 16.6HIJ |
| Glyphosate | TOM102 | 20.0GHIJ |
| Glyphosate | TOM47 | 15.0HIJ |
| Quinclorac | TOM66 | 6.6H |
| Quinclorac | TOM129 | 3.3H |
| Quinclorac | TOM77 | 10.0EGH |
| Quinclorac | TOM410 | 15.0GH |
| Quinclorac | TOM63 | 8.3EGH |
| Aminocyclopyrachlor | TOM27 | 13.3FG |
| Aminocyclopyrachlor | TOM74 | 8.3FG |
| Aminocyclopyrachlor | TOM54 | 20.0EFG |
| Aminocyclopyrachlor | TOM78 | 10.0FG |
| Aminocyclopyrachlor | TOM29 | 6.6FG |
| Aminocyclopyrachlor | TOM44 | 5.0FG |
| Aminocyclopyrachlor | TOM129 | 5.0G |
| Aminocyclopyrachlor | TOM103 | 18.3EFG |
| Picloram | TOM17 | 20.0HIJ |
| Picloram | TOM47 | 13.3J |
| Aminopyralid | TOM76 | 10.0F |
| Aminopyralid | TOM84 | 18.3EF |

* Different letter indicates the mean injury for the accessions are significantly different at $p < 0.05$.

The effect of 2,4-D was significantly different on all the accessions with a $p$-value < 0.0001 for injury. TOM17 showed the least injury, with 4.5%. It belongs to the species *pennellii*, with unusual morphology and tolerance to extreme stress such as salinity, making it one of the most abiotic stress taxa of tomato [28]. The leaves of *pennellii* are very thick as compared to the cultivated tomato; leaf analysis of a five-week-old *pennellii* plant reveals it has 0.94% of its dry weight in epicuticular lipids, whereas *Solanum lycopersicum* (cultivated tomato) only has 0.16% of the leaf dry weight in epicuticular lipids [29]. This thick cuticle of TOM17 may have reduced the penetration of 2,4-D through leaves, thus leading to lesser injury [23]. The other two accessions with injury of 5% or less were TOM83 and TOM56. TOM83 was

reported to show moderate resistance to *Pepino mosaic virus* (PepMV), a highly contagious disease in greenhouse tomatoes, whereas TOM56 has resistance to black mold, a disease of ripe tomato fruit caused by *Alternaria alternate*, and this disease resistance trait from TOM56 has been bred into cultivated tomatoes [30,31].

According to Atkinson and Urwin (2012), there is a significant overlap in signaling and response pathways to different abiotic and biotic stresses, which consists of cellular redox status, hormones, reactive oxygen species, protein kinase cascades, and calcium gradients as common elements [32]. The overlap in signaling pathways is associated with cross-tolerance phenomena in which plants also develop resistance to other biotic or abiotic stresses [33]. Thus, as expected, the tolerance of these accessions to abiotic and biotic stresses may also lead to 2,4 D tolerance.

For dicamba herbicide, there were eleven accessions with less than 20% injury, ranging from 6.5 to 20%. Accessions with the least injury included TOM17, TOM13, and TOM1, with 6.5, 7.3, and 8.5% injury, respectively on 28 DAT. Interestingly, TOM17 showed the least injury (4.5%) to 2,4-D, and was tolerant to dicamba too. Another accession that showed low injury to dicamba was TOM13 (7.3%), which belongs to the species *pimpinellifolium*, and is more specifically used to combat biotic stresses such as tomato yellow leaf curl virus, *Botrytis cinerea*, and *Fusarium oxysporum* f.sp. *lycopersici* [34,35]. Additionally, TOM13 has been found to have some drought-tolerant traits [36]. TOM1, which is among the accessions that showed least injury to dicamba, is reported to be partially resistant to genetically distinct strains of *Clavibacter michiganensis* subsp. *michiganensis*, which causes bacterial canker, a serious pathogen causing significant yield losses in tomatoes grown in humid conditions [37]. Resistance from TOM1 was recovered in lines from a BC2S4 inbred backcross (IBC) population in both greenhouse and field trials, and the same line can be tested for 2,4 D tolerance in future studies.

Glyphosate-tolerant accessions showed injury ranging from 3 to 20%, with a total of nine accessions falling under this category. Among the nine accessions, TOM60 (3.3%), TOM61 (10%), and TOM46 (10%) showed the lowest injury. TOM60 is reported to be resistant to two insects, two-spotted spider mite (*Tetranychus urticae*) and sliverleaf whitefly (*Bemisia tabaci*), based on egg numbers using leaf disc and damage score bioassays [38]. TOM61 belongs to *Solanum chilense*, a drought-tolerant species, and it is five times more tolerant to wilting compared to cultivated tomatoes. These wild taxa of tomato have longer primary roots, and more extensive secondary root systems which make them drought-tolerant species [39]. TOM46 has a tolerance to higher temperatures [28]. Tomatoes are extremely sensitive to quinclorac; however, in the current study, we found five tolerant accessions with injury of less than 20%. The lowest injury of 3.3% was observed for TOM129. The other two accessions with low injury were TOM66 and TOM63. TOM129 belongs to *S. lycopersicum* var. *cerasiforme*, which is a cherry tomato biotype. Ciccarese et al. [40] reported that accessions belonging to *S. lycopersicum* var. *cerasiforme* species show high tolerance to powdery mildew caused by *Oidium lycopersici*, and a single recessive gene was reported to be responsible for tolerance. Moreover, Cillo et al. [41] showed that *S. lycopersicum* var. *cerasiformeis* were tolerant to cucumber mosaic virus. TOM63 is a *pennellii* accession like TOM17, while TOM66 belongs to *S. chmielewskii*, which are found to be moderately resistant to fungal pathogen *Oidium neolycopersici*. The production of reactive oxygen species (ROS) and peroxidase activity during the infection of *O. neolycopersici* is associated with the activation of defense responses in genotypes, thus indicating the presence of this defense system in TOM66 [42].

For picloram herbicide, only two tolerant accessions were identified. The accessions TOM17 (also tolerant to 2,4-D) and TOM47 (same species as TOM129, *S. lycopersicum* var. *cerasiforme*) were tolerant to picloram.

For aminocyclopyrachlor, eight accessions showed injury of less than 20%, of which TOM44 and TOM129 showed the least injury of 5%. Both accessions belong to *S. lycopersicum* var. *cerasiforme*, as discussed previously, and TOM129 is also tolerant to quinclorac herbicide.

For aminopyralid herbicide, TOM76 and TOM84 showed injury of less than 20%. Recently, a major QTL (known as *stm9*) on chromosome 9 was identified in TOM76, which is associated with the maintenance of shoot turgor under root chilling. Root chilling (6 °C) induces the rapid onset of water stress by impeding water movement from roots to shoots. TOM76 responds to such changes by closing stomata and maintaining shoot turgor, while *S. lycopersicum* fails to close stomata and wilts [43]. TOM84 is similarly found to be tolerant to low temperatures ranging from 2 to 4 °C [28].

Most of the tolerant accessions belong to the same species, *Lycopersicon*, as commonly grown tomato cultivars (Figure 3), thus, indicating the ease of crossing between commercial cultivars and tolerant lines in breeding programs.

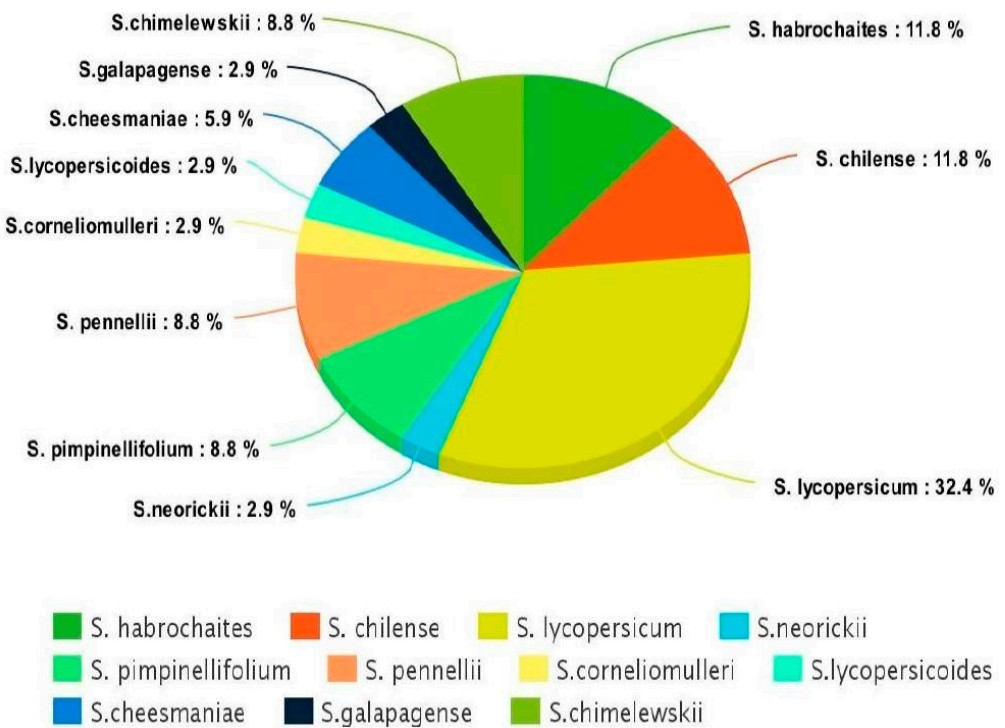

**Figure 3.** Distribution of tolerant accessions discovered in the study under different species (cultivated and wild relatives) of tomato.

The other two large groups used in this study were *Solanum habrochaites and S. chilense* (11.8%). *Solanum habrochaites* is a source of various forms of biotic stress, and has recently been reported to be a potential source of resistance against *Bactericera cockerelli* (Hemiptera: Triozidae) and *Candidatus Liberibacter solanacearum* [44]. Similarly, *S. chilense* is found to be tolerant to low temperatures (abiotic stress) in that none of the plants showed any wilting or visible injury when exposed to 4 and 2 °C, which is atypical for tomato species [45]. The other two groups that are widely studied and used frequently in abiotic stress breeding programs are *S. pimpinellifolium* and *S. Pennellii*. Bolger et al. [46] successfully sequenced the genome of the *S. pennellii*, and numerous QTLs have been identified for salt tolerance in these species. A linkage map of crosses between *Solanum lycopersicum* and *Solanum pimpinellifolium* displays the genomic locations of resistant gene analogs and candidate resistance/defense–response ESTs [47].

In summary, the current study identified tomato genotypes that have a higher natural tolerance to various herbicides. These genotypes can be used as raw genetic materials for the development of herbicide-tolerant tomato varieties that can produce higher yields by minimizing yield loss due to herbicide drift and weeds. However, while working on the pipeline to develop herbicide-tolerant/resistant varieties, breeders should also keep in mind the specific risks associated with these varieties. The two major factors associated with

herbicide tolerance are environmental and health concerns, and the breeders should conduct a detailed need/market assessment before fully investing in elite varieties development.

## 4. Conclusions

The study identifies tomato genotypes tolerant to commonly drifted herbicides in tomato production systems. Almost all the herbicide-tolerant lines were also tolerant to other biotic/abiotic stresses. Tomato breeders can use the lines identified in this study to breed new tomato varieties with herbicide tolerance. These lines can be used as an important genetic resource in tomato breeding programs. Additionally, with the help of molecular biology techniques and information available on the tomato genome, breeders can identify QTLs responsible for herbicide tolerance, thus aiding in marker-assisted breeding. Once successful tomato varieties are developed that have herbicide tolerance and good yield and quality potential, they can be made available to tomato growers to help combat herbicide drift-related issues and herbicide sensitivity. Information regarding the tolerant lines and QTLs responsible for herbicide tolerance would be submitted to tomato genetic databases such as the Tomato Genetic Resource Center at UC Davis and made available to researchers and breeders worldwide. Future research should focus on understanding the mechanism of herbicide tolerance in the genotypes with higher herbicide tolerance.

**Supplementary Materials:** The following supporting information can be downloaded at: https://www.mdpi.com/article/10.3390/horticulturae9121354/s1, Table S1: List of all accessions used in this study along with their species and place of origin. These accessions are tolerant to various abiotic/biotic stress, it includes both wild and cultivated tomatoes.

**Author Contributions:** Conceptualization, T.-M.T. and G.S.; methodology, G.S., validation, T.-M.T.; formal analysis, G.S. and S.S. (Swati Shrestha); investigation, G.S.; data curation, S.S. (Sanju Shrestha); writing—review and editing, G.S., S.S. (Swati Shrestha), and S.S. (Sanju Shrestha). All authors have read and agreed to the published version of the manuscript.

**Funding:** Funding for this project was provided by the Specialty Crop Block Grant sponsored by the Mississippi Department of Agriculture and Commerce/U.S. Department of Agriculture—Agriculture Marketing Service, and is based upon work that is supported by the National Institute of Food and Agriculture, U.S. Department of Agriculture, Hatch project under accession number 230100.

**Data Availability Statement:** The data presented in the study are available upon request from the corresponding author. The data is not publicly available due to future analytical studies.

**Acknowledgments:** The authors want to acknowledge Ziming Yue and Shandrea Stallworth for their support in helping with the experiment.

**Conflicts of Interest:** The authors declare no conflict of interest.

## Nomenclature

Glyphosate: N-(phosphonomethyl) glycine, 2,4-D: (2,4-dichlorophenoxy) acetic acid, dicamba: 3,6-dichloro-2-methoxybenzoic acid, quinclorac: 3,7-dichloro-8-quinolinecarboxylic acid, aminocyclopyrachlor: 6-amino-5-chloro-2-cyclopropyl-4-pyrimidinecarboxylic acid, tomato, *Solanum lycopersicum*.

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
