# Peer review of "Screening and Selection for Herbicide Tolerance among Diverse Tomato Germplasms"

_horticulturae, doi:10.3390/horticulturae9121354_

Round 1

Reviewer 1 Report

Comments and Suggestions for Authors

A very interesting study bringing new knowledge in the field of crops tolerant to herbicides that has been elaborated in high-quality. This new trend in weed control has been already used in a number of crops. I really appreciate the quality of work and an interesting idea. I have a few minor specific comments related to the work.

1. Introduction

·         Line 48 – please use SI units of measurement. „pounds“ should be replaced, there is no need to mention them.

·         Lines 55: Please state scientific names of the weeds: „purple nutsedge, large crabgrass, and palmer amaranth“

·         Line 52, 66 – Currency „dollars“ should be used consistently throughout the text – even expressed by words or by symbol. It needs to be unified.

2. Materials and Methods

·         Lines 123, 132 – please use SI units of measurement

·         Lines 139-142 – What does „X“ stand for?

3. Results and Discussion

·         Expand the discussion. Please pay attention to the risks associated with growing herbicide-tolerant crops

3.1. Figures, Tables and Schemes

·         Do not list as a separate chapter

·         Figures and Tables should be incorporated to the text in chapter "3. Results and Discussion"

·         The quality of Figure 3 needs to be improved

·         Table 3. – Please round the values off using the same amount of decimal spaces – 2nd and 3rd column

·         Table 4. – Can it be alligned (shifted) to the right?

Reviewer 2 Report

Comments and Suggestions for Authors

1) Lines 57-68: The impact of herbicide usage must be mentioned. You can simply said that “herbicide usage can impact to environment [https://doi.org/10.3390/ijerph20032738], soil microorganisms [ https://doi.org/10.3389/fmicb.2023.1285445], and human health [https://doi.org/10.3390/ijerph19063198]” Please see these papers.

2) Line 134: “At 4-leaf stage” how many days after sowing? Please explain.

3) Lines 139-142: “Drift rates were selected based on previous studies” Please explain more details why these rates were used? The readers can understand here without finding the previous studies.

4) Line 263: “3.1. Figures, Tables and Schemes” must be deleted or try to define the sub-topic, like 3.1 and 3.2.

5) Figure 1 should be enlarged the font size.

6) Figure 2: Why the line was not at 50%?

Comments on the Quality of English Language

-

Round 2

Reviewer 2 Report

Comments and Suggestions for Authors

Thank you for the revised manuscript. Although some comments have been improve, a few comments remain un-improved.

-Lines 57-68: The impact of herbicide usage must be mentioned. You can simply said that “herbicide usage can impact to environment [https://doi.org/10.3390/ijerph20032738], soil microorganisms [ https://doi.org/10.3389/fmicb.2023.1285445], and human health [https://doi.org/10.3390/ijerph19063198]” Please see these papers.

Comments on the Quality of English Language

-

Author Response

Comments addressed in the attached file.
